# Simulation of Extreme Ultraviolet Radiation and Conversion Efficiency of Lithium Plasma in a Wide Range of Plasma Situations

**Xiangdong Li [1],\*, Frank B. Rosmej [2,3],\* and Zhanbin Chen [4]**

[1] State Key Laboratory of High Field Laser Physics and CAS Center for Excellence in Ultra-Intense Laser Science, Shanghai Institute of Optics and Fine Mechanics (SIOM), Chinese Academy of Sciences (CAS), Shanghai 201800, China

[2] Faculty of Sciences and Engineering, Sorbonne University, Case 128, Place Jussieu, F-75252 Paris, France

[3] Ecole Polytechnique, LULI, Physique Atomique dans les Plasmas Denses, Route de Saclay, F-91128 Palaiseau, France

[4] School of Science, Hunan University of Technology, Zhuzhou 412007, China; chenzhanbin008@qq.com

\* Correspondence: xiangdong_li@siom.ac.cn (X.L.); frank.rosmej@sorbonne-universite.fr (F.B.R.)

**Abstract:** Based on the detailed term accounting approach, the relationship between extreme ultraviolet conversion efficiency and plasma conditions, which range from 5 to 200 eV for plasma temperature and from $4.63 \times 10^{17}$ to $4.63 \times 10^{22}$ cm$^{-3}$ for plasma density, is studied for lithium plasmas through spectral simulations involving very extended atomic configurations, including a benchmark set of autoionizing states. The theoretical limit of the EUV conversion efficiency and its dependence on sustained plasma time are given for different plasma densities. The present study provides the necessary understanding of EUV formation from the perspective of atomic physics and also provides useful knowledge for improving EUV conversion efficiency with different technologies.

**Keywords:** plasma spectra; lithium plasma; extreme ultraviolet (EUV); EUV conversion efficiency; multi-configuration Dirac–Fock (MCDF) method





## 1. Introduction

Extreme ultraviolet lithography (EUVL) is a state-of-the-art technology used to produce semiconductor chips beyond the 10 nm node [1] with extremely short wavelength light sources (13.5 ± 0.135 nm). Among the critical technologies involved in EUVL, the development of an extreme violet light source is the most important one. Currently, the generation of intense EUV light for commercial applications mainly relies on plasma radiation. Therefore, laser- and discharge-produced plasmas (LPPs and DPPs, respectively) have become the two mainstream technologies for the development of EUV light sources [2,3]. ASML [4], one of the rare suppliers of EUV lithography machines in the world, adopts the LPP solution.

Xenon, tin, and lithium are considered the three most promising EUV-emitting elements [5–14]. This is because some special charge states of these three elements have rich and strong EUV emission channels compared to other elements in the periodic table. Thus, these three elements have the potential to be used as efficient sources of EUV radiation when proper plasma conditions are achieved. However, due to their different energy level structures and material properties, they show obvious differences in their spectra and EUV conversion efficiency (EUVCE).

In the framework of LPP technology, tin stands out and has become the only element that is commercially used due to its relatively high EUVCE. However, tin plasmas radiate very broad and strong spectra outside in-band EUV radiation. This makes it very difficult to achieve nearly monochromatic EUV light. Therefore, complex and expensive optical systems, such as multilayer film reflection and imaging optical systems, have to be used to narrow the bandwidth, but this increases both the cost and the technical difficulty.

Lithium plasma has a very concise (or simple) spectral structure; its hydrogen-like 1s–2p transition is considered to be the main channel to provide EUV radiation (13.5 nm) with good monochromaticity. This is an important advantage over other candidates. Unfortunately, in many LPP experiments, the EUVCE of lithium plasma is lower than that of tin plasma under the same laser-interaction conditions and is also difficult to improve. This is why LPP technology for lithium has not yet been commercially applied.

In order to improve the EUVCE of lithium plasma, some researchers have proposed some technical solutions, including a tin–lithium mixture and the use of DPP technology [15]. However, these attempts did not make significant progress in EUVCE. Today, scientists still do not have clear knowledge or a method of how to effectively improve EUVCE. The evidence and phenomena provided by experiments are important but not sufficient to help understand all the key details of EUV radiation. Since the theoretical simulation of plasma spectra is an irreplaceable way to display the spatial–temporal evolution of EUV radiation, theory will play an indispensable role in the development of an optimized EUV light source.

In this article, the detailed term accounting (DTA) approach combined with local thermal equilibrium (LTE) conditions are used to simulate the spectra of lithium plasma over a broad range of temperatures and densities. Accordingly, the variations in EUVCE with plasma temperature and density are obtained. It is shown that at a given plasma density, there is a maximum EUVCE with the change in plasma temperature, and this maximum increases with plasma density. If plasma opacity is included, an optimal condition exists at which the EUVCE reaches a maximum. It is realized that the sustained plasma time under this optimal plasma condition for both LPPs and DPPs is the key factor in improving the EUVCE. The changes in the EUVCE and the sustained plasma time provide useful knowledge about the EUV radiation generated by lithium plasma from an atomic physics point of view.

## 2. Atomic Structure Data

The atomic structure data used in this work are produced using the multi-configuration Dirac–Fock (MCDF) method, which is widely used and has been described extensively in the literature [16]. Hence, we shall not provide details regarding the MCDF theory here. However, since the total plasma energy and the total plasma radiation energy are necessary for calculating the EUVCE, it is required, in principle, to include all energy levels and transitions in the calculations. This is usually quite cumbersome, and one has to choose as many configurations as possible so that the calculation of the total plasma energy is close to its true value. Therefore, we list the selection of configurations in the article as follows.

For H-like lithium, configurations with a principle quantum number of n ≤ 25 are included, which is very close to the continuum.

For He-like lithium, configurations with single- and double-electron excitation are included. For configurations of single-electron excitation, we limit the maximal principal quantum number, n, to 25, as for H-like lithium. For configurations of double-electron excitation, we limit one excited electron to n ≤ 8 and the other excited electron to n ≤ 25. This results in more than twice the ionization energy of the first order. We list the choice of configurations for the He-like lithium as follows:

1*s*(2), 1*s* 2*l*, 1*s* 3*l*, . . .. . .1*s* 25*l*;
2*l*(2), 2*l* 3*l*, 2*l* 4*l*, . . .. . .2*l* 25*l*;
3*l*(2), 3*l* 4*l*, 3*l* 5*l*, . . .. . .3*l* 25*l*;
4*l*(2), 4*l* 5*l*, 4*l* 6*l*, . . .. . .4*l* 25*l*;
5*l*(2), 5*l* 6*l*, 5*l* 7*l*, . . .. . .5*l* 25*l*;
6*l*(2), 6*l* 7*l*, 6*l* 8*l*, . . .. . .6*l* 25*l*;
7*l*(2), 7*l* 8*l*, 7*l* 9*l*, . . .. . .7*l* 25*l*;
8*l*(2), 8*l* 9*l*, 8*l* 10*l*, . . .. . .8*l* 25*l*.

The symbol n*l*(m) means all the possible configurations for m electrons in the orbitals with a principal quantum number of n, and n*l* means all the possible configurations for

one electron in the orbitals with a principal quantum number of n. For neutral lithium, the choice of configurations is similar to that for the He-like ions. Configurations with three electron excitations are not included because the excitation energies far exceed the ground state ionization energies. We list the choice of configurations for neutral lithium as follows:

$1s(2)\,2l$, $1s(2)\,3l$, $1s(2)\,4l$, .......$1s(2)\,25l$;
$1s\,2l(2)$, $1s\,2l\,3l$, $1s\,2l\,4l$, ......$1s\,2l\,25l$;
$1s\,3l(2)$, $1s\,3l\,4l$, $1s\,3l\,5l$, ......$1s\,3l\,25l$;
$1s\,4l(2)$, $1s\,4l\,5l$, $1s\,4l\,6l$, ......$1s\,4l\,25l$;
$1s\,5l(2)$, $1s\,5l\,6l$, $1s\,5l\,7l$, ......$1s\,5l\,25l$;
$1s\,6l(2)$, $1s\,6l\,7l$, $1s\,6l\,8l$, ......$1s\,6l\,25l$;
$1s\,7l(2)$, $1s\,7l\,8l$, $1s\,7l\,9l$, ......$1s\,7l\,25l$;
$1s\,8l(2)$, $1s\,8l\,9l$, $1s\,8l\,10l$, ......$1s\,8l\,25l$.

The symbols are the same as for He-like lithium. All electric dipole transitions involving the above configurations are calculated using the flexible atomic code (FAC) [17], which is a sophisticated atomic code based on the MCDF method.

## 3. Charge State Distribution and Spectral Simulation

Because the LTE plasma condition is assumed, the charge state distribution for a plasma can be conveniently solved through the Saha equation.

$$\frac{N_{i+1}}{N_i} = \frac{Z_e Z_{i+1}}{Z_i N_e} e^{-\frac{\phi_i}{kT}}, \tag{1}$$

where $N_i$ and $N_{i+1}$ are the population densities of charge state $i$ and $i+1$, respectively. $N_e$ is the plasma electron density, $\phi_i$ is the ionization potential of charge state $i$. $Z_e$, $Z_i$, and $Z_{i+1}$ are the partition functions for a free electron, charge state $i$, and charge state $i+1$, respectively, which are given by

$$Z_e = 2\left(\frac{2\pi m_e KT}{h^2}\right)^{\frac{3}{2}}, \tag{2}$$

$$Z_i = \sum_j g_{ij} e^{-\frac{E_{ij}}{kT}}, \tag{3}$$

where $m_e$ is the mass of an electron, $h$ is the Planck constant, $g_{ij} = 2J_{ij} + 1$ is the statical weight for level $j$ of charge state $i$, and $E_{ij}$ is the energy of level $j$ of charge state $i$ relative to its ground state energy. For the partition function, $Z_i$, the summation runs over the chosen levels, which are extensive.

The population density of level $j$ of charge state $i$ under LTE is given by

$$N_{ij} = g_{ij} \frac{N_i}{Z_i} e^{-\frac{E_{ij}}{kT}}. \tag{4}$$

Equation (1) is solved with the constraint of particle conservation,

$$N = \sum_{ij} N_{ij}, \tag{5}$$

and charge conservation.

The power of the line emission for a transition from an upper level, $l$, to a lower level, $l'$, in charge state $i$ is calculated by

$$P_{il'l}(h\nu) = N_{il} A_{il'l} E_{il'l} S(h\nu), \tag{6}$$

where $A_{il'l}$ is the spontaneous transition probability from an upper level, $l$, to a lower level, $l'$, in charge state $i$, and $S(h\nu)$ is the line-shape function, which is assumed to be of the Doppler type

$$S(h\nu) = \frac{\sqrt{ln2}}{\sqrt{\pi}\Gamma}e^{ln2(h\nu-h\nu_0)^2/\Gamma^2},$$ (7)

where $E_{il'l} = h\nu_0$ is the transition energy from level $l$ to $l'$ in charge state $i$. The Doppler width is given by

$$2\Gamma = 7.716 \times 10^{-5}\left(\frac{kT}{A}\right)^{\frac{1}{2}}(h\nu_0),$$ (8)

where $kT$ is the plasma temperature in eV, $A$ is the atomic weight, and $h\nu_0$ is the transition energy in eV.

Finally, the total plasma spectral power can be calculated from the summation of all these line emissions.

$$P(h\nu) = \sum_{il'l} N_{il}A_{il'l}E_{il'l}S(h\nu)$$ (9)

## 4. Total Plasma Energy and EUV Conversion Efficiency

In order to calculate the EUVCE under given plasma conditions, the absorbed energy and the radiation energy of the plasma are necessary. The energy absorbed by the plasma is converted into three parts—ionization energy, excitation energy, and thermal energy. So, the absorbed energy for plasma in thermal equilibrium with a temperature $T$ and a particle density $N$ can be calculated as follows:

The total thermal energy is given by

$$E_{thermal} = N_e \times \frac{3}{2}kT + N \times \frac{3}{2}kT,$$ (10)

the total excitation energy is calculated by

$$E_{excitation} = \sum_{ij} N_{ij}E_{ij},$$ (11)

and the total ionization energy is

$$E_{ionization} = \sum_i N_iE_i,$$ (12)

where $E_i$ is the energy needed to ionize the neutral state to charge state $i$. So, the total absorbed energy of the plasma with temperature $T$ and particle density $N$ is

$$E_{total} = E_{thermal} + E_{excitation} + E_{ionization}.$$ (13)

The total radiation power for the bound–bound emission is

$$P_{total} = \sum_{il'l} N_{il}A_{il'l}E_{il'l},$$ (14)

where only electric dipole transitions are considered. The total in-band EUV radiation power is given by

$$P_{in-band} = \sum_{il'l} N_{il}A_{il'l}E_{il'l},$$ (15)

where only in-band electric dipole transitions are included.

The total continuum radiation power is calculated by the Bremsstrahlung, including free–free and free–bound radiation, as

$$P_{f-f} = \int_0^\infty \frac{32}{3} \left(\frac{\pi}{3}\right)^{\frac{1}{2}} \frac{e^4}{m_e^2 c^3} \left(\frac{R_y}{kT}\right)^{\frac{1}{2}} \overline{Z} n_e^2 \exp\left(-\frac{h\nu}{kT}\right) d(h\nu) \cong 9.6 \times 10^{-14} (eV^{\frac{1}{2}} cm^3 s^{-1}) \overline{Z} n_e^2 \sqrt{kT}$$

$$P_{f-b} = \int_{E_{\zeta-1,m}}^\infty \frac{64}{3} \left(\frac{\pi}{3}\right)^{\frac{1}{2}} \frac{e^4}{m_e^2 c^3} \left(\frac{E_{\zeta-1,m}}{kT}\right)^{\frac{1}{2}} \zeta N_\zeta n_e \left(\frac{1}{n_p}\right)^3 \exp\left(-\frac{h\nu}{kT}\right)(1 - P_{\zeta,m}) d(h\nu)$$

(16)

$$\cong 5.22 \times 10^{-14} \left(\frac{E_{\zeta-1,m}}{kT}\right)^{\frac{1}{2}} \zeta N_\zeta n_e \left(\frac{1}{n_p}\right)^3 (1 - P_{\zeta,m}) kT exp\left(-\frac{E_{\zeta-1,m}}{kT}\right)$$

(17)

where $\overline{Z}$ is the average charge state of the plasma, $\zeta$ is the charge of an ionic state, $N_\zeta$ is the number density of charge state $\zeta$, $E_{\zeta-1,m}$ is the binding energy of a recombining electron after its recombination to state $m$ from charge state $\zeta$, $P_{\zeta,m}$ is the population probability of the final bound state. Details about Equations (16) and (17) are given in reference [18]. In this paper, for free–bound radiation, only recombination to the ground state is considered because the radiation power is inversely proportional to the cube of the principal quantum number, $n_p$. If the sustained plasma time under thermal equilibrium is $\Delta t$, then the EUVCE can be calculated by

$$EUVCE = \frac{P_{in-band} \times \Delta t}{E_{total} + P_{total} \times \Delta t + P_{f-f} \times \Delta t + P_{f-b} \times \Delta t} \times 100$$

(18)

It is clear that when $\Delta t$ is close to infinity, the EUVCE reaches its maximum.

## 5. Results and Discussion

This paper aims to provide some necessary knowledge concerning EUV emissions from lithium plasma from the perspective of atomic physics so as to find and understand some key points to improve the EUVCE. At the same time, this paper also hopes to realize whether lithium plasma has a limit to its EVUCE and what this limit is. Therefore, the radiation spectrum of lithium plasma is simulated over a temperature range of 5 to 200 eV and a density range of 0.534 g/cm$^3$ to 0.00000534 g/cm$^3$, which covers from solid density ($N = 4.63 \times 10^{22}$ cm$^{-3}$) to gas density ($N = 4.63 \times 10^{17}$ cm$^{-3}$) and is the plasma region of interest in the study of EUV light sources. It should be noticed that the plasma density extends to a relative lower value of $4.63 \times 10^{17}$ cm$^{-3}$, at which the assumption of the Saha equation probably does not hold well. We introduce the local thermal equilibrium (LTE) criterion of Griem [19] as follows:

$$n_e \geq 9 \times 10^{17} \left(\frac{E_{12}}{E_{ionization}}\right)^3 \left(\frac{kT}{E_{ionization}}\right) cm^{-3}$$

(19)

where $E_{ionization}$ is the ground-state ionization energy of a H-like ion, $E_{12}$ is the energy gap between states 1 and 2 in a H-like ion, and T is the plasma temperature. If the energy gap, $E_{12}$, extends to the continuum, such as 200 eV, and the plasma temperature is 100 eV, the critical electron density for which H-like Li will reach thermal equilibrium will be about $3.2 \times 10^{18}$cm$^{-3}$. If an average ionization degree of two is used, the critical ion density for H-like Li to reach thermal equilibrium is about $1.6 \times 10^{18}$cm$^{-3}$. So, the LTE condition holds under most of our plasma conditions, and even at a density of $4.63 \times 10^{17}$ cm$^{-3}$, LTE can still be used to produce approximate results.

Figure 1 illustrates the charge state distribution with changes in plasma temperature and density. This contains important evolution information about the absolute and relative density distributions for different charge states, which is essential for understanding the formation of the EUVs and optimizing the EUVCE. It shows that as plasma density decreases, the average degree of ionization increases at the same plasma temperature. That means it is easier to obtain a higher ionization state at lower plasma densities because three-body recombination becomes less effective at lower densities.

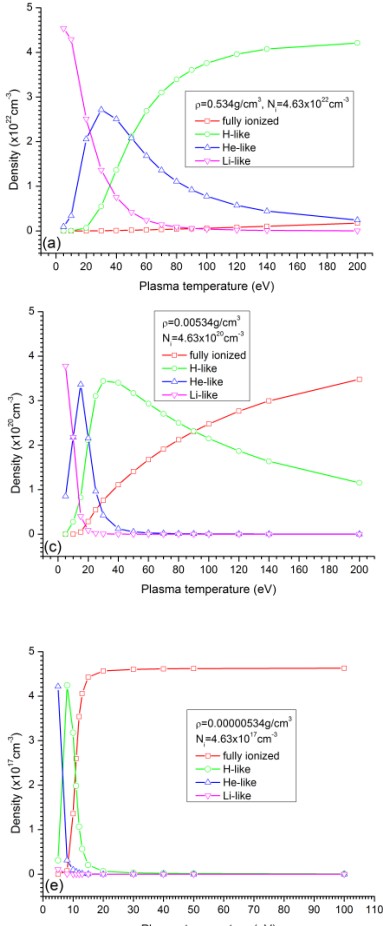
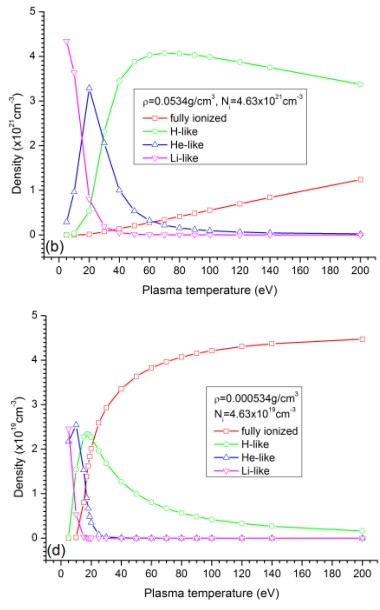

**Figure 1.** Charge state distribution with changes in plasma temperature and density for lithium plasma. Figures (**a**–**e**) correspond to a decrease in plasma density.

The plasma conditions under which the maximal distribution is reached for each charge state are also important for studying EUV radiation. As indicated in Figure 1, a maximal density distribution always exists for each charge state at a given plasma density. This maximal density decreases as the plasma density decreases (see columns 4 and 5 in Table 1). However, the proportion of this maximum distribution in the plasma does not decrease monotonically with plasma density (see columns 7 and 8 in Table 1). As an example, Table 1 gives a comparison for the He-like and H-like lithium densities when they achieve maximums at different plasma densities.

**Table 1.** Comparison of maximum He-like and H-like lithium densities for plasma densities of interest. The corresponding temperature is the temperature at which the He-like and H-like lithium states achieve their maximum distributions at the corresponding plasma densities. Proportion (%) is the ratio of the maximum density of He-like or H-like Li to the plasma density at the corresponding temperature.

| Plasma Density $(cm^{-3})$ | Corresponding Temperature (eV) | | Maximal Density $(cm^{-3})$ | | Proportion (%) | |
|---|---|---|---|---|---|---|
| | He-like Li | H-like Li | He-like Li | H-like Li | He-like Li | H-like Li |
| $4.63 \times 10^{22}$ | 30 | 200 | $2.716 \times 10^{22}$ | $4.21 \times 10^{22}$ | 58.6 | 90.9 |
| $4.63 \times 10^{21}$ | 20 | 70 | $3.289 \times 10^{21}$ | $4.07 \times 10^{21}$ | 71.03 | 87.9 |
| $4.63 \times 10^{20}$ | 15 | 30 | $3.36 \times 10^{20}$ | $3.44 \times 10^{20}$ | 72.57 | 74.3 |
| $4.63 \times 10^{19}$ | 10 | 17 | $2.54 \times 10^{19}$ | $2.34 \times 10^{19}$ | 54.8 | 50.5 |
| $4.63 \times 10^{17}$ | 5 | 8 | $4.217 \times 10^{17}$ | $4.24 \times 10^{17}$ | 91 | 91.5 |

It is clear that the absolute density of a charge state determines the radiation intensity of this charge state, and the proportion of a charge state (its relative distribution in plasma) determines the radiation efficiency of this charge state. The higher the absolute density of a charge state, the greater its radiation intensity. The higher the relative density (proportion) of a charge state, the higher its radiation efficiency. As we can see from Table 1, the proportion of a certain charge state does not monotonically increase or decrease with its absolute density. For He-like lithium, a proportion of 72.57% corresponds to an absolute density of He-like ions of $3.36 \times 10^{20}$ cm$^{-3}$. But at a plasma density of $4.63 \times 10^{22}$ cm$^{-3}$, a proportion of He-like ions of 58.6% corresponds to an absolute density of He-like ions of $2.716 \times 10^{22}$ cm$^{-3}$. An increase in the absolute density does not lead to an increase in the relative density (proportion). Equally, an increase in radiation intensity does not necessarily lead to an increase in radiation conversion efficiency. Since in developing an EUV light source, both EUV intensity and efficiency need to be improved, a balance between the absolute and relative density needs to be struck (also known as optimization).

Figure 2 shows the radiation spectrum of lithium plasma at different temperatures and densities. Obviously, the spectrum of lithium plasma is quite simple, and it provides in-band EUV radiation with fairly good monochromaticity. At the same time, the in-band EUV emission (13.5 ± 0.135 nm) from lithium plasma corresponds practically to the strongest line in the spectrum. Regarding the other relatively strong lines, the line at 10.8 nm is the Lyman-γ line of H-like Li; the line at 11.4 nm is a mixture of Lyman-β with lines from He-like Li; and the line at 19.9 nm is from He-like Li, respectively. But their intensities are far lower than the in-band EUV radiation. These properties represent the most significant advantages of EUV light sources.

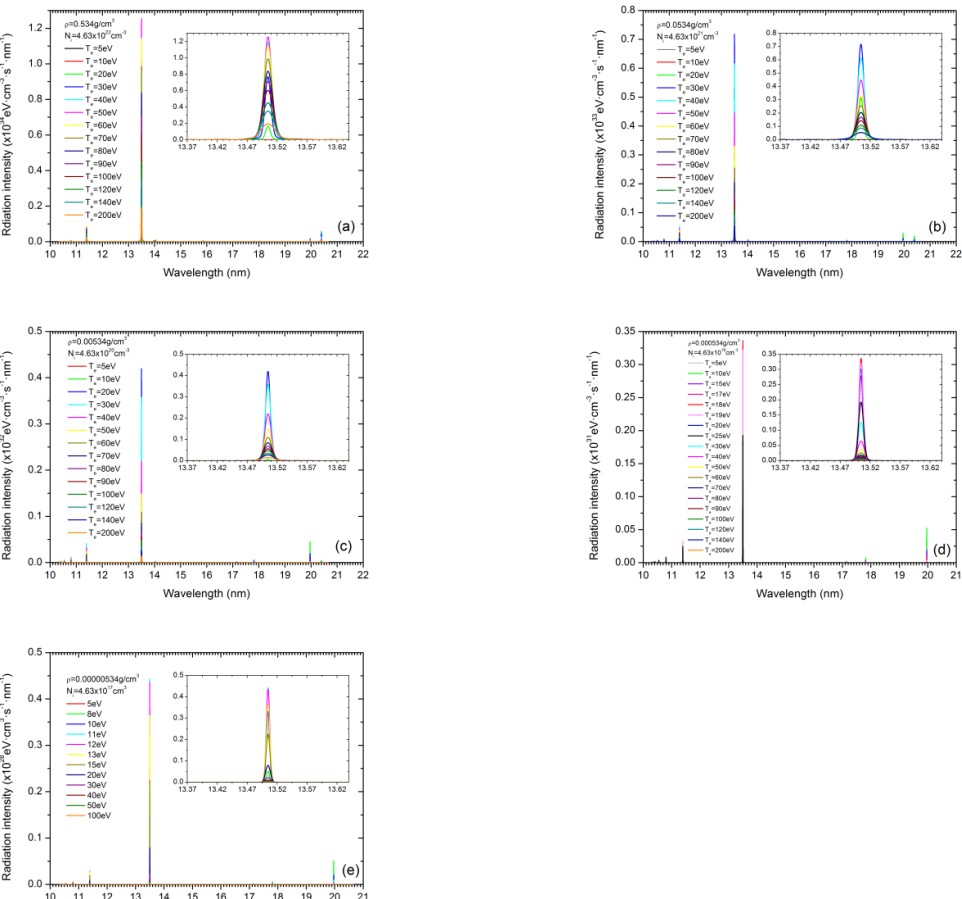

**Figure 2.** Predicted EUV radiation spectra for lithium plasma over a range of densities and temperatures. Figures (**a**–**e**) correspond to decreasing plasma ion density (see insets). The frame inside each figure shows the in-band EUV line spectrum at 13.5 ± 0.135 nm, which is used as a light source for lithography.

According to Figure 2, for a given plasma density, the EUV radiation power can achieve a maximal value at a certain temperature, and this maximal radiation power increases with plasma density. So, increasing the plasma density is one necessary way to improve the intensity of EUV radiation. However, this improvement in EUV intensity is more complex due to self-absorption [20]. In fact, its opacity increases with plasma density, and incorporating opacity into the EUVCE calculation lead to a balance between radiation and absorption. Therefore, there must be an optimal point for temperature and density to achieve the maximum EUV emission power. Here, we focus on optimization from atomic physics and plasma spectroscopic points of view related to the maximum EUVCE in lithium plasma. It should be mentioned that only Doppler broadening is considered in the spectral simulations. But this does not affect the calculation of the total radiation energy since the total radiation energy is independent of spectral line broadening. However, line broadening has an impact on calculating the total in-band EUV energy, as it can make the line span out of the in-band EUV area. A report by Lorenzen and coworkers [21] showed that the pressure broadening of the Lyman-$\alpha$ line of Li$^{+2}$ at a plasma electron density of $N_e \approx 10^{22}$ cm$^{-3}$ (corresponding to $N \approx 5 \times 10^{21}$ cm$^{-3}$ if the average degree of ionization is taken to be two) is less than 0.1 nm, which was not out of the in-band EUV range of 0.27 nm. If linear extrapolation is applied to Lorezen's work, an approximate 2.7 nm broadening for Lyman-$\alpha$ at a density of $N \approx 4.63 \times 10^{22}$ cm$^{-3}$ is obtained (see Figure 3a). So, for plasma densities near the solid density, the total energy of in-band EUV radiation was overestimated. However, as the density decreases, the spectral broadening will further decrease. Thus, spectral broadening also has little influence on the calculation of the total in-band EUV energy if the plasma density is lower than the solid density of lithium.

At any fixed plasma density, there is a maximum EUV radiation power at a specific plasma temperature. Figure 3 shows the strongest in-band EUV emissions and the corresponding temperatures at which a transition from solid to gas density takes place. It is very interesting to see that the in-band EUV radiation originates from both H-like (red line) and He-like (green line) lithium. The contributions of these two charge states to the total in-band EUV radiation change with the plasma density. When the plasma density is high, the main contribution to the in-band EUV radiation is from the He-like ions (see Figure 3a in the solid plasma density range; only a 6% contribution to the total EUV comes from H-like ions). With the decrease in plasma density, the hydrogen-like ions gradually became the main contributors to the EUV radiation. When the lithium particle density is less than 0.0000534 g/cm$^3$ ($4.63 \times 10^{18}$ cm$^{-3}$), almost all of the EUV radiation comes from the hydrogen-like ions. It is clear that for the He-like lithium, the in-band EUV radiation is not from the single-electron excitation configurations ($1s^1nl^1-1s^2$) but from the double-electron excitation configurations ($nl^1ml'^1-1s^1nl^1$ and $nl^1ml'^1-1s^1ml'^1$, where n and m > 1). So, in-band EUV radiation for He-like lithium requires higher temperatures to produce double-electron excitation. However, when the plasma density is low and the plasma temperature is high, the degree of ionization increases so that the number of He-like ion states becomes small. It can be seen that in a density range of $4.63 \times 10^{17}$ to $4.63 \times 10^{19}$ cm$^{-3}$, the temperature for the lithium plasma to reach the maximum intensity of the in-band EUV emission is about 10–20 eV. In such a situation, H-like ions constitute the main charge state. Although the He-like ions emit strong EUV radiation at high densities, this EUV radiation basically cannot escape from the plasma due to its strong opacity. Therefore, the EUV radiation that can be observed in the experiment probably comes from H-like ions.

Figure 4 illustrates the variations in the theoretical EUVCE with plasma temperature for various plasma densities and sustained plasma times. One can clearly see that the EUVCE depends principally on sustained plasma time. For any given plasma parameters, maintaining a long sustained plasma time is an important way to improve the emission conversion efficiency. At a given plasma density and sustained plasma time, there is a plasma temperature at which the EUVCE reaches its maximum. When the sustained plasma time goes to infinity (see Equation (18) at $\Delta t = \infty$), the EUVCE approaches its maximum for a given plasma density. Therefore, at any given plasma density, there

exists a maximum EUVCE, and this maximum increases with decreasing plasma density. For instance, the maximum EUVCE for $\rho = 0.534$ g/cm$^3$ is 40.68% at a temperature of 30 eV. However, the maximum is about 87.98% for $\rho = 0.0000053$ g/cm$^3$ at a temperature $T = 11$ eV. Consequently, low-density plasmas have a higher EUVCE (please note that the EUVCE is not equal to the EUV intensity or power). The reason behind this is the fact that as the plasma temperature and density increase, the radiation power of the Bremsstrahlung increases nonlinearly, causing most of the energy absorbed by the plasma to be converted into continuous radiation. This leads to a decrease in the conversion efficiency. It is also interesting to see that although the maximum EUVCE is larger at lower plasma densities, longer sustained times are needed to reach the same EUVCE compared to higher densities. For example, a sustained time of 10 ns for plasma at a density of $\rho = 0.53$ g/cm$^3$ can reach a maximum EUVCE value of up to 27.96%. But for a plasma at a density of $\rho = 0.0000053$ g/cm$^3$, the maximum EUVCE value is only about 2.1% for the same sustained time of 10 ns. This means that for low-density plasmas, a longer sustained time is needed to reach the same EUVCE as a higher-density plasma. In fact, it is difficult to maintain a plasma state for a long time for both the LPP and DPP schemes. On the other hand, in order to balance radiation efficiency and radiation intensity, it should be feasible to increase both the EUVCE and EUV intensity for higher-density plasma.

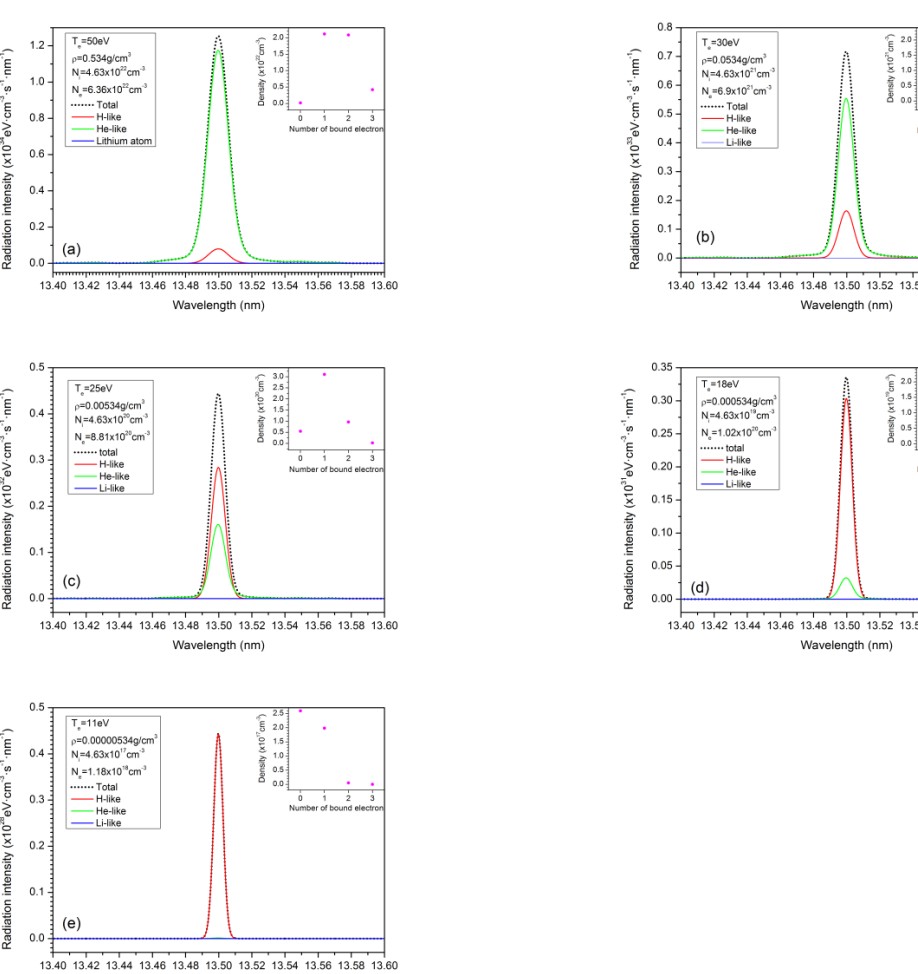

**Figure 3.** Predicted in-band EUV emission intensities versus wavelength for several plasma densities of interest. The plasma electron temperatures used for each graph (**a**–**e**) correspond to the temperatures at which maximum intensities were obtained in Figure 2, graphs (**a**–**e**). Here, the black dot lines correspond to the total in-band EUV radiation. The contributions from different charge states are shown by different colored lines (see inset). The charge state distributions for the corresponding plasma conditions are plotted in the inner frame using pink solid-circle curves.

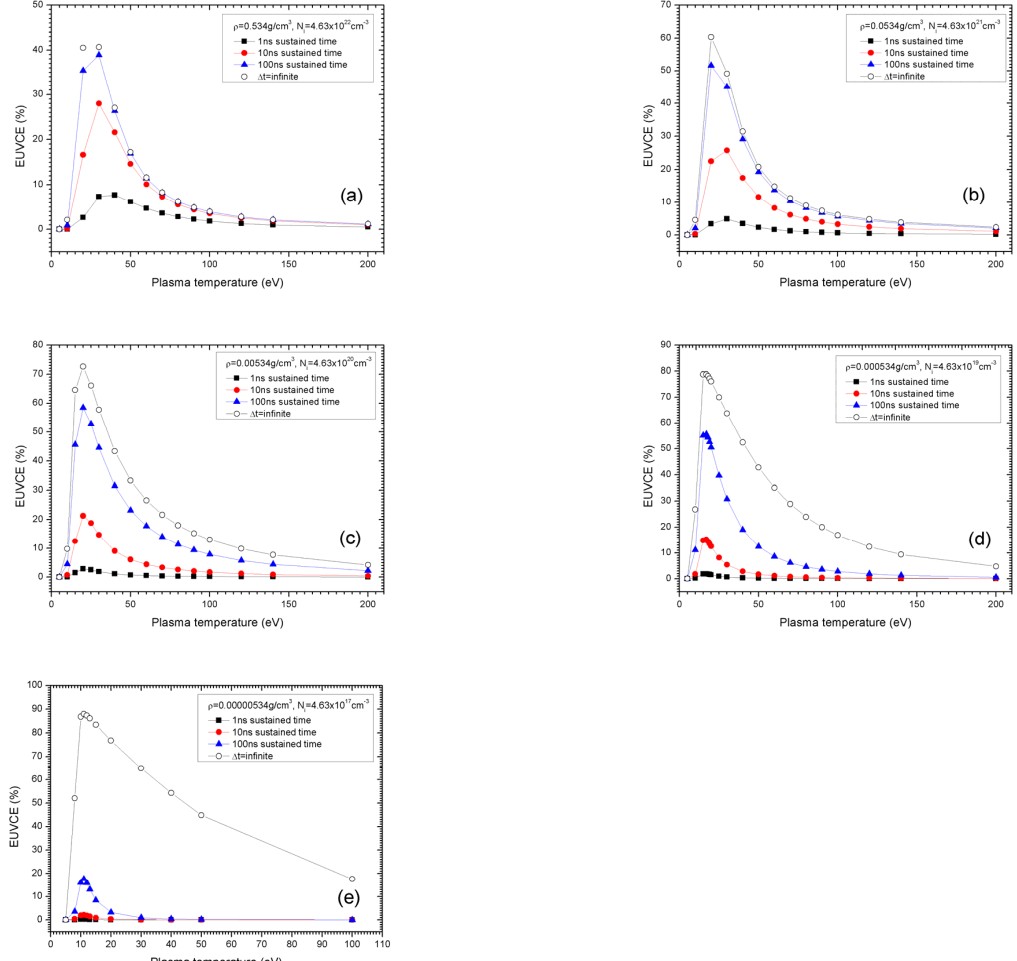

**Figure 4.** The EUV conversion efficiency (EUVCE) under different plasma conditions and different sustained plasma times. Figures (**a**–**e**) correspond to a decrease in plasma density.

Comparing Figures 2 and 4, it is observed that the temperature to reach the maximum EUV radiation power does not correspond to the temperature that maximizes the EUVCE value at a given plasma density.

## 6. Conclusions

In order to minimize the impact of a limited atomic structure on the EUV conversion efficiency, a very extended atomic structure configuration set has been used. Particular attention has been paid to autoionizing states, including up to very high $nln'l'$-quantum numbers ($n \leq 8$, $l \leq 7$, $n' \leq 25$, $l' \leq 24$), that may be considered a benchmark set with respect to atomic structure. The potential EUVCE may reach values up to 87% at low densities if the sustained plasma time is sufficiently long. However, when the sustained plasma time is lower than 1ns, the theoretical EUVCE decreases strongly. Therefore, efficient plasma confinement to realize a long sustained time is critical for EUVCE optimization.

EUV radiation intensity (or power) and EUV conversion efficiency are two different physical quantities. Radiation intensity increases with density, but conversion efficiency does not. Taking into account the sustained time and the plasma opacity, it is necessary to find a balance between EUVCE and radiation intensity to find a compromise for EUVCE that is technically and commercially acceptable. These compromises in EUVCE optimization are quite challenging and might be further illuminated via radiation hydrodynamic simulation, which will be a subject for future research.

**Author Contributions:** Conceptualization, X.L. and F.B.R.; methodology, X.L. and F.B.R.; software, X.L.; validation, X.L., F.B.R. and Z.C.; formal analysis, X.L. and F.B.R.; investigation, X.L F.B.R. and Z.C; resources, X.L. and F.B.R.; data curation, X.L., F.B.R. and Z.C; writing—original draft preparation, X.L.; writing—review and editing, F.B.R. and Z.C.; visualization, X.L.; supervision, X.L. and F.B.R.; project administration, X.L. and F.B.R.; funding acquisition, X.L. and F.B.R. All authors have read and agreed to the published version of the manuscript.

**Funding:** This research was funded by the NSFC under Grant Nos. 12074395 and 11374315 and the invited scientist program of CNRS at Ecole Polytechnique, Palaiseau, France.

**Data Availability Statement:** All data are available by contact the email address xiangdong_li@siom.ac.cn (due to privacy restrictions).

**Conflicts of Interest:** The authors declare no conflict of interest.

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
