# Peer review of "Simulation of Extreme Ultraviolet Radiation and Conversion Efficiency of Lithium Plasma in a Wide Range of Plasma Situations"

_atoms, doi:10.3390/atoms12030016_

Round 1
Reviewer 1 Report
Comments and Suggestions for Authors
The paper presents new simulations of EUV radiation and conversion efficiencies for lithium plasmas over a wide range of plasma conditions. The simulations undertaken by the authors are based on the detailed term-accounting approach and using this approach they undertake spectral simulations involving very extended atomic configurations that includes very high auto-ionizing states. They consider that this study provides new insights into understanding EUV formation from an atomic physics perspective while also providing useful information for improving the EUV efficiency conversion I consider that the results they provide can be of significant interest to the EUCL community as they attempt to develop new EUV light sources. This study might also find interest in the area of liquid-lithium based first-wall materials for magnetic confinement fusion, in particular when simulating disruptions using high-power lasers. In general, the paper is well written and clear. However, before publication, the paper requires minor corrections to the text and figures as well as some clarifications. For instance, to reach an extended audience outside the EUCL community, the authors could provide some more detailed explanations in certain parts of their work. I provide a list for the authors. Once these are completed, I can recommend the paper for publication.
Specific Comments
Abstract:
The abstract describes fairly well the work presented. However, some small details could be added to enhance it.
1) Plasma condition: some indication of the plasma limits used in this study should be provided.
Main text:
Page 1: lines 26 to 28: It is difficult to understand this phrase. Maybe “Currently, the generation of intense EUV with commercial applications is very dependent on plasma emission conditions.”
Page 1: line 39: Please modify to “tin stands out as the only one that is …”.
Page 2, line 48: Please modify to “the EUVCE of lithium plasma is lower than that for tin plasma under the same laser-interaction conditions and is also difficult to improve.”
Page 2, lines 85 & 86: Please modify to “It is required, in principle, to include all energy levels and transitions in the calculations.”
Page 4, line 143: Please define Ail’l for readers who may not be fully familiar line emissions.
Page 4, equation (17): Please define here in the text Eζ-1,m, ζ, Nζ , Pζ,m for readers who may not be fully familiar with these terms.
Page 4, equation (17): Please change 5,22 to 5.22.
Page 5, lines 188 and 189: Please modify to “This paper aims to provide some necessary knowledge concerning EUV emissions from lithium plasma from ...”
Page 5, lines 195 to 198: It would be useful to allow readers to qualify this statement by providing a generally accepted lower density limit for LTE conditions, maybe > 1018 cm-3.
Page 5, lines 207 to 209: The authors write, “As indicated in Fig. 2, a maximum density distribution always exists for each charge state at a given plasma density”. Are you not discussing Fig. 1 here?
Page 5, Table 1, Caption: Please modify to “Comparison of maximum He-like and H-like densities for plasma densities of interest. Corresponding temperature is the temperature at which the He-like and H-like lithium states achieve their maximum distributions at the corresponding plasma density. Proportion (%) is the ratio of Maximum density of He-like or H-like Li to Plasma density at Corresponding temperature.”
Page 5, lines 218 to 221: Please use “determines” rather than decide”.
Figure 1: The authors should be consistent with the labels in all 5 graphs presented here. For instance, use either “Plasma electron temperature” or “Plasma temperature” and “Density” or “Ion density”. As the fonts are very small it would be better to label (a) as (x1022 cm-3), (b) as (x1021 cm-3), etc. Also, the labelling of both the x and y axis should be consistent. For instance, 0, 20, 40, … 180, 200 or 0, 50, 100, 150 and 200. The same with the y axis. Also, on the y axis please remove “-5.0x1020” for (b) and “-1x1017” from (e) as they make no sense. Finally, use “fully ionized” rather than “bare nuclear”. Also, what do the points represent on the graphs? Plasma temperatures for which estimates have been made? I guess then that the lines are provided to guide the reader’s eye.
Figure 2: Again, the authors should be consistent with the labels in all 5 graphs presented here. For instance, please use either “Radiation intensity” or “Intensity”. Also correct “Wave length (nm)” to “Wavelength (nm)” in (a). Also, as the fonts are very small, it would be better to label (a) as (x1034 eV · cm-3 s-1 · nm-1), (b) as (x1032 eV · cm-3 s-1 · nm-1), etc. In (c) s-3 should be corrected to s-1.
It is also very difficult (impossible) to distinguish between the multiple curves in all figures plotted here. In one case, 18 curves appear, many of which overlap. I suggest reducing the number of curves in each graph so that the reader can interpret what the data is saying.
With regard to the caption, maybe something like the following will improve understanding. “Predicted EUV radiation spectra for lithium plasma over a range of densities and temperatures. Figures (a) to (e) correspond to decreasing plasma ion density (see insets). The frame inside each figure shows the in-band EUV line spectrum at 13.5+/-0.135 nm which is used as a light source for lithography.”
Additionally, for completeness, the authors could identify, in one graph or in the text, the other lines in these figures, i.e., at 10.8 nm, 11.39 nm and 19.9 nm.
Page 7, line 254: Please modify to “…. since the total radiation energy is independent of spectral line broadening”.
Page 7, lines 254 to 257: This phrase is difficult to understand. Maybe it should be “At the same time, a paper by Lorenzen and coworkers [20] showed that the pressure broadening of Lyman-lines in dense Li+2 plasma is less than 0.1 nm, which is within the in-band EUV range of 0.27 nm”. Also, “dense” should be quantified here.
Figure 3: Again, as the font are very small in all graphs. It would be easier to read label (a) as (x1034 eV · cm-3 s-1 · nm-1), (b) as (1032 eV · cm-3 s-1 · nm-1), etc. Please do the same for the inner graphs. Also, the x label should be “Number of bound electrons”. Maybe increase the font sizes to aid readers.
The text in the caption should be modified to explain better the origins of these graphs. For example, “Predicted in-band EUV emission intensities versus wavelength for several plasma densities of interest. The plasma electron temperatures used for each graph (a) to (e) correspond to the temperatures at which maximum intensities were obtained in Figure 2, graphs (a) to (e). Here, the black dot-dot lines correspond to total in-band EUV radiation. The contributions from different charge states are shown by different color lines (see in-set). The charge state distributions for the corresponding plasma conditions are plotted in the inner frame using pink solid-circle curves.”
Page 8, lines 268 to 270: It would help readers who are not fully familiar with such things if the authors provided a short explanation why in-band EUV radiation originates from H-like and He-like lithium.
Page 9, line 296: Please modify to “… various plasma densities and sustained plasma times.“ In other parts, it is better to use “sustained plasma time”.
Page 9, line 297: Please replace “seriously depends on …” with “depends principally on …”.
Page 9, lines 299 and 300 and Figure 4: In Figure 4, the plasma temperature at which maximum EUVCE (%) is achieved appears to depend on Dt, e.g., in graph (a) the maximum of EUVCE (%) moves towards lower temperatures as Dt becomes longer whereas in graph (d) the maximum of EUVCE (%) appears to move towards higher temperatures as Dt becomes longer. Can the authors comment on this in the text.
Page 9, line 301: Should this not be eq. (18) rather than eq. (17)?
Page 9, lines 302 to 303: It is difficult to understand the phrase “Therefore, at any given plasma density there exists a maximum EUVCE and this maximum increases with decreasing plasma density”.
Figure 4: The limits used in the x-axis of (b) and (e) should the same as those used in (a), (c) and (d). Also, negative values of EUVCE (%) should be eliminated. It would be easier for readers to compare all graphs if the y-axis limits were the same for all graphs, e.g., maximum 100%.
Comments on the Quality of English LanguageThere are numerous small errors in the English text that need attention. Some are noted in Comments and Suggestions. Also, the authors should review agreements between singular and plural nouns and verbs. There are numerous incorrect matches.
Author Response
Thanks very much for the professional review from the referee. We have carefully read our manuscript and revised all the English writing as the referee point out. Also, we modified all the figures formation according to the referee’s suggestions. All these revisions are highlighted with the yellow color. As for the questions provided by the referee, I give my answer one by one as follows.
Question 1. Also, what do the points represent on the graphs of Fig. 1? Plasma temperatures for which estimates have been made? I guess then that the lines are provided to guide the reader’s eye.
Answer 1. In Fig. 1, all graphs are presented by line+symbol formation to show the Charge state distribution with the change of plasma temperature and density. The points (or the symbols) correspond to the temperature we used in the calculation.
Question 2. It is also very difficult (impossible) to distinguish between the multiple curves in all figures plotted in Fig. 2. In one case, 18 curves appear, many of which overlap. I suggest reducing the number of curves in each graph so that the reader can interpret what the data is saying.
Answer 2. Yes it is difficult to distinguish the multiple curves on the long scale coordinate (from 10 to 22nm) because of so many temperatures are chosen in the calculations. So, we provide an inside frame for each figure to show the clear details in the in-band EUV area. The reason to provide the long scale coordinate spectra lays in two aspects. One is to show that under a given plasma density there exist the maximum radiation for the in-band EUV with the variation of plasma temperature. Other is to show the spectral properties that represent the most significant advantages among the candidates of EUV light source that are good monochromaticity, high intensity of the in-band EUV radiation and the simple spectral structure. These two properties can be clearly seen in the long scale coordinate, although some of the spectra overlapping with each other. We also add some explanation on the manuscript to other relative strong line as the referee suggested.
Qustion 3. It is difficult to understand the phrase “Therefore, at any given plasma density there exists a maximum EUVCE and this maximum increases with decreasing plasma density”.
Answer 3. For example, at given plasma density 0.534g/cm3 (4.63x1022cm-3), the maximum EUVCE is 40.7% when the sustained plasma time goes to infinite (see Fig. 4a). When plasma density decrease to 0.0534 g/cm3 (4.63x1021cm-3) the maximum EUVCE at the infinite sustained plasma time increase to 60.3% and then increase to 87.9% as the plasma density decrease to 0.00000534g/cm3 (4.63x1017cm-3).
Question 4. In Figure 4, the plasma temperature at which maximum EUVCE (%) is achieved appears to depend on Dt, e.g., in graph (a) the maximum of EUVCE (%) moves towards lower temperatures as Dt becomes longer whereas in graph (d) the maximum of EUVCE (%) appears to move towards higher temperatures as Dt becomes longer. Can the authors comment on this in the text.
Answer 4. As for this question, I carefully checked the data of EUVCE. I found it is a false impression. I listed the plasma temperature at which maximum EUVCE (%) is achieved as follows.
Δt |
plasma temperature at which maximum EUVCE (%) is achieved |
||||
0.534g/cm3 |
0.0534g/cm3 |
0.00534g/cm3 |
0.000534g/cm3 |
0.00000534g/cm3 |
|
1nm |
40eV |
30eV |
20eV |
17eV |
11eV |
10nm |
30eV |
30eV |
20eV |
17eV |
11eV |
100nm |
30eV |
20eV |
20eV |
17eV |
11eV |
infinite |
30eV |
20eV |
20eV |
17eV |
11eV |
As can be seen in above table, when small temperature step is used in the lower density the plasma temperature at which maximum EUVCE (%) is achieved have nothing to do with the sustained plasma time. For the dense plasma, because the temperature step 10eV is used, so both the maximum EUVCE and the corresponding temperature are inaccurate. It cannot come to the result that the plasma temperature at which maximum EUVCE (%) is achieved decrease with the sustained plasma time. Detailed calculation with small temperature step is needed.
Reviewer 2 Report
Comments and Suggestions for Authors
This paper presents the result of the analysis of EUV emission in the 13.5 nm band from lithium plasma. Assuming the local thermodynamic equilibrium (LTE) state of plasma and considering simple energy balance conversion efficiency to the EUV emission is discussed, which is of interest to the readers of the atom journal. Thus, the paper should be published after minor amendments.
The representation of the spectrum in Fig. 2 and 3 should be improved, with some additional statements for the structure and broadening mechanism for the emission lines.
In Fig. 2, lines for more than ten different temperatures overwrap, and each line is hardly identified.
In contrast to the emission from H-like Li through a single 1s-2p line, emission from He-like Li should arise from several satellite lines. However, the structure in Fig.3 looks very similar in the case of both H-like and He-like lines dominating the spectrum. Furthermore, the authors only take Doppler broadening into account. The effects of other broadening mechanisms should be mentioned depending on the density and temperature. In addition, identification of satellite lines that have significant contributions may be helpful. If the authors use a log plot, the contribution of the continuum may be clarified.
Author Response
Thanks very much for the professional review from the referee. We have carefully read our manuscript and revised all the English writing as the referee point out. Also, we modified all the figures formation according to the referee’s suggestions. All these revisions are highlighted with the yellow color. In addition, the referee also has two questions.
Question 1. Fig 2 and Fig 3 are hardly identified. It need to be improved.
Answer 1. Yes it is difficult to distinguish the multiple curves on the long scale coordinate (from 10 to 22nm) because of so many temperatures are chosen in the calculations. So, we provide an inside frame for each figure to show the clear details in the in-band EUV area. The reason to provide the long scale coordinate spectra lays in two aspects. One is to show that under a given plasma density there exist the maximum radiation for the in-band EUV with the variation of plasma temperature. Other is to show the spectral properties that represent the most significant advantages among the candidates of EUV light source that are good monochromaticity, high intensity of the in-band EUV radiation and the simple spectral structure. These two aspects can be clearly seen in the long scale coordinate, although some of the spectra overlapping with each other. We also add some explanation on the manuscript to other relative strong line as the referee suggested (see line 255-258 highlighted).
Question 2. In contrast to the emission from H-like Li through a single 1s-2p line, emission from He-like Li should arise from several satellite lines. However, the structure in Fig.3 looks very similar in the case of both H-like and He-like lines dominating the spectrum. Furthermore, the authors only take Doppler broadening into account. The effects of other broadening mechanisms should be mentioned depending on the density and temperature. In addition, identification of satellite lines that have significant contributions may be helpful. If the authors use a log plot, the contribution of the continuum may be clarified.
Answer 2. Yes the in-band EUV emission from the He-like Li is come from the decay of a double electron excitation state nl1ml’1-1s1nl1 and nl1ml’1-1s1ml’1, where n and m >1. We have given some explanation about this on the line 300-303 of page 9. (Highlighted with yellow color)
Yes, only Doppler broadening is considered in this paper. But, some analysis for other broadening mechanism are added in the line 270-282 of the paper 8. (Highlighted with Yellow color)